# Use of Phil Embolic Agent for Bleeding in Non-Neurological Interventions

**DOI:** 10.3390/jcm10040701

**Published:** 2021-02-11

**Authors:** Pierleone Lucatelli, Mario Corona, Leonardo Teodoli, Piergiorgio Nardis, Alessandro Cannavale, Bianca Rocco, Claudio Trobiani, Stefano Cipollari, Simone Zilahi de Gyurgyokai, Mario Bezzi, Carlo Catalano

**Affiliations:** Interventional Radiology Section of Department of Radiological, Oncological, and Anatomopathological, Sciences of Policlinico Umberto I of Rome, Sapienza University of Rome, Rome 00161, Italy; pierleone.lucatelli@gmail.com (P.L.); mario.corona@uniroma1.it (M.C.); piergiorgio.nardis@uniroma1.it (P.N.); alessandro.cannavale@hotmail.com (A.C.); bianca_rocco@hotmail.com (B.R.); claudio.trobiani@gmail.com (C.T.); stefanocipollari@gmail.com (S.C.); simonezilahi@hotmail.it (S.Z.d.G.); mario.bezzi@gmail.com (M.B.); carlo.catalano@uniroma1.it (C.C.)

**Keywords:** transcatheter embolization, liquid embolic agent, non-neurologic intervention, bleeding

## Abstract

Objective: To evaluate the safety and efficacy of the Phil liquid embolic agent in non-neurological embolization procedures. M&M: Thirty-five patients with a mean age of 62.5 years underwent percutaneous embolization using Phil for the treatment of visceral arterial bleedings in 20/35 patients (including three gluteal, one bladder, two superior mesenteric, three epigastric, one deep femoral, five internal iliac, four intercostal, and one lingual arteries), splanchnic pseudoaneurysms in 11/35 patients (including three hepatic, five splenic, and three renal arteries), pancreatic bleeding metastasis in 1/35 patient, and gastric bleeding varices in 3/35 patients. Phil is composed of a non-adhesive copolymer dissolved in DMSO (Anhydrous Dimethyl Sulfoxide) with different viscosity. Procedures were performed slowly under continuous fluoroscopic guidance to avoid embolization of non-target vessels. Results: Clinical success was obtained with a single intervention in 34 cases (97.15%), while a repeated procedure was required in one case (2.85%). No technical complications nor non-target embolization occurred. A case of post-embolic syndrome was noted (2.85%) in one patient. DMSO administration-related pain was successfully controlled by medical therapy. Conclusion: Phil can be considered a safe and effective embolic agent for the treatment of non-neurologic bleeding.

## 1. Introduction

Hemodynamically relevant abdominal bleeding is always an acute or subacute emergency requiring prompt treatment. Trans-catheter embolization has evolved into a mainstay in the management of acute bleeding, being introduced in several guidelines as a therapeutic tool to be employed in stable patients [1,2,3,4,5].

Transcatheter embolization is a minimally invasive procedure that combines excellent technical and clinical outcome with a low risk of complications [2,5]. With the latest development of embolic agents and devices (e.g., miniaturization of microcatheters), embolization is the treatment of choice for many vascular pathologies (in neurological and non-neurological districts, such as spontaneous hematomas, pseudoaneurysms, arterio-venous malformations or fistulae and hyper-vascular tumors), especially when patients have massive bleeding, are in poor clinical condition, and/or are unfit for surgery such as elderly fragile patients [6,7].

Various agents, including polyvinyl alcohol particles (PVA), acrylic polymers, coils, microspheres, alcohol, and n-butylcyanoacrylate (NBCA) have been used for selective arterial embolization procedures in many vascular pathologies without reaching satisfactory clinical results [8]. The ideal embolic agent is still not available, especially in the emergency setting; it should yield a rapid and effective embolization, reach and fill the targeted distal artery branches without migration, be easy to prepare, highly radiopaque, controllable during the administration, biocompatible, and cost-effective.

Phil (Precipitating Hydrophobic Injectable Liquid) is the trade name of a non-adhesive liquid embolic agent comprised of a biocompatible polymer dissolved in dimethyl sulfoxide (DMSO) solvent. Its reported clinical applications, according to published series, are within the neurovascular field (brain arterio-venous malformations (BAVM) and aneurysms) [9,10,11,12]. There are currently no studies on the efficacy of Phil on non-neurological application; however, a recent study on animal models reported that the efficacy of Phil in non-neurological procedures is comparable to that of similar embolic agents [13].

The aim of this study is to report the safety and efficacy of Phil in the treatment of acute and subacute hemorrhage outside of the neurovascular area.

## 2. Materials and Methods

Between December 2019 and December 2020, 133 patients with acute or subacute hemorrhage were referred to our IR unit. Of these, thirty-five patients (21 males, 14 females) with a mean age of 62.25 years (range 20–89 years) underwent emergency percutaneous embolization using Phil. Application includes acute and subacute setting (such as arterial or portal varices bleedings), localized both at abdomen (such as gastrointestinal and urinary bleeding) and at peripheral level (such as intramuscular spontaneous hematomas and post-traumatic bleeding). Indications were: visceral arterial bleedings (*n* = 20, three gluteal artery, one bladder bleeding, two superior mesenteric artery, three epigastric artery, one deep femoral artery, five internal iliac artery, four intercostal artery, and one lingual artery), splanchnic pseudoaneurysm (*n* = 11, three hepatic artery, five splenic artery, and three renal artery), pancreatic hypervascular bleeding metastasis (*n* = 1), and gastric bleeding varices (*n* = 3) (Figure 1 and Figure 2).

All patients underwent pre-treatment evaluation with CT-angiography (CTA) and/or MR-angiography (MRA).

All patients were informed of the intended treatment and possible complications and a written consent was obtained before the procedure.

Procedures were performed in the angio-suite by an interventional radiologist (experience > 10 years) and started under local anesthesia.

Super-selective catheterization of the branches feeding the bleeding/pseudoaneurysm was performed using a DMSO-compatible 2.7 Fr. microcatheter (Progreat Microcatheter 2.7 Fr, Terumo Medical Corporation, Tokyo, Japan).

Phil injection was done very slowly under fluoroscopic guidance to avoid occlusion or embolization of non-target vessels.

When bleeding occurred at high blood flow, we considered using metal coils in addition to Phil. Mean procedural time was recorded, it was calculated from the making of the arterial vascular access to the removing of the vascular introducer. We also recorded the amount of embolic agent used in each patient to achieve the arrest of bleeding.

Local anesthesia has always been performed before the percutaneous puncture for vascular access and before administering DMSO; the anesthesiologist has always been advised by the interventional radiologist in order to provide a deep sedation administering propofol to control DMSO-related pain.

Follow-up was performed with clinical evaluation and imaging modalities (CTA and/or MRA) on a regular basis at 1–6–12 months.

### PHIL System^®^

The Phil liquid embolic device (Precipitating Hydrophobic Injectable Liquid, MicroVention Terumo, Tustin, CA, USA) is composed of a nonadhesive copolymer (polylactide-co-glycolide and polyhydroxyethylmethacrylate) dissolved in Anhydrous Dimethyl Sulfoxide (DMSO) with an iodine component (triiodophenol) covalently bound to the copolymer, causing radiopacity [14,15]. Phil is supplied in pre-filled 1 mL sterile syringes [15].

DMSO is a widely used commercial solvent derived from trees as a byproduct from the production of paper [16]. In the body, DMSO rapidly oxidizes to dimethyl sulfone (methlysulfonylmethane-MSM) and dimethyl sulfide. Both DMSO and MSM are soluble in both oil and water-based liquids. However, dimethyl sulfide is hydrophobic and tends to be insoluble in water and soluble in oil-based liquids. Clearance of DMSO and MSM is achieved by both excretion in the urine and feces and by elimination through the lungs and skin in the form of dimethyl sulfide.

Phil solidifies through a process of precipitation initiated when it encounters an aqueous solution (e.g., blood, body fluids, normal saline, water), and the solvent DMSO rapidly diffuses out of the polymer mass, causing in-situ precipitation of a soft radiopaque polymeric embolus. Complete precipitation occurs within 3 to 10 min [15].

The microcatheter must be primed with the DMSO solvent to fill its dead space. Afterwards, Phil is first slowly injected to displace the DMSO, and then the injection is continued at a slow steady rate (recommended rate 0.16 mL/min). 

The iodine component allows visualization of Phil under fluoroscopy, with no risk of microcatheter blockage, at the same time minimizing artifacts at follow-up imaging studies and facilitating staged procedure or repeated treatments. It is compatible with surgical resection with no reported hazards related to sparking/combustion and no tattoo effect in the treatment of superficial malformation.

Phil is available in a range of liquid viscosities (25%, 30% and 35%): lower viscosities, achieved by reducing the polymer/dimethyl sulfoxide ratio, are indicated for the embolization of arteriovenous malformations where depth penetration in small diameter vessels is required.

The primary objective of this study was to evaluate the safety of the Phil in non-neurological interventions.

The following parameters were taken into account for safety evaluation assessment: post-procedural pain; non-target embolization; post-embolic syndrome (PES), assessed by occurrence of fever, nausea and pain; infection following embolization; and allergic reactions to the embolic material.

Efficacy was evaluated by means of technical success and clinical success. Technical success was defined as the immediate arrest of bleeding in acute hemorrhage and complete devascularization of varices in cases of bleeding from portal hypertension. Clinical success was defined based on the need for repeated treatment.

## 3. Results

Phil injection was correctly performed in all cases and no technical complications were recorded. Technical success was obtained with a single intervention in 97.15% of the cases, while a repeated procedure was required in 2.85% of the cases. Specifically, all arterial bleedings were successfully treated with a single procedure, while a case of portal hypertension with associated splenorenal shunt and hemorrhagic gastric varices required a second embolization procedure due to the inability to achieve complete devascularization of the varices during the first procedure.

The average consumption of Phil per single patient was 1.09 mL, in particular, in nine procedures (20%) it was necessary to use more than 1 mL of embolizing agent without ever exceeding 3 mL of product. 

In two patients, metal coils were used in addition to Phil to reduce blood flow during embolization; particularly we used detachable coils Concerto Detachable Coil System ev3, (Medtronic, Minneapolis, MN, USA) in one patient and detachable coils RUBY Coil System (Alameda, CA, USA) in the other patient, both of variable diameter and length.

At the end of the embolization, the DMSO-compatible microcatheter was easily removed in all cases by applying a gently traction on it; no cases of catheter entrapment occurred. 

Mean procedural time was 57 min.

Pain caused by DMSO injection was noted in all procedures and was controlled by medical therapy (deep sedation) administered during the procedure, in accordance with the anesthesiologist.

PES was found in only one patient (2.55%), which was one of the 15 patients (6.67%) undergoing embolizations for abdominal visceral hemorrhage. The patient presented with abdominal pain, nausea, vomiting, and fever immediately following embolization of multiple intrapancreatic bleeding renal cancer metastases infiltrating the duodenal wall. The symptoms were promptly limited with medical therapy based on analgesics and antiemetics.

At post-procedural follow-up imaging performed with CT, MRI at 1, 6, or 12 months, there were no signs of significant non-target embolization. No clinical signs and symptoms of no-target embolization were reported at follow-up. 

Procedures were well tolerated by all patients without evidence of systemic or local toxicity. No pain or symptoms related to inflammatory response, such as allergic reactions, were observed during the entire follow-up. No infections of the embolic material were reported in the follow-up, and no abscesses or post-procedural fluid collections were observed at follow-up imaging.

## 4. Discussion

The results of our study show that Phil has a level of safety comparable to other embolizing agents on the market, even in non-neurological use [17]. As for clinical success, this was obtained in 97.15% of cases treated, thus achieving a good result, in line with what has been reported in other studies for similar embolizing agents [18,19,20]. In our experience, there were no technical failures in the use of this material, which is easy to use and does not require dedicated training.

Embolization procedures can be performed using different agents such as polyvinyl alcohol particles, acrylic polymers, coils, gelfoam, microspheres, alcohol, and n-butylcyanoacrylate. These materials, however, may be associated with high toxicity and strong inflammatory reaction [21,22].

The Phil liquid embolic agent was designed primarily for interventional neuroradiological procedures [5,12]. Its non-adhesive properties make its use more predictable as compared with other liquid agents, and it represents a valid tool in the pre-surgical treatment of neurologic malformations [12]. 

Based on these characteristics, we consider using this embolizing material for the treatment of non-neurological bleeding, both visceral and peripheral, including visceral pseudoaneurysms, bleeding metastases, and bleeding gastric varices, as well as spontaneous hematomas and peripheral post-traumatic bleeding. At the time of submission, the use of Phil in non-neurological interventions is not reported in the literature. One study on the use of Phil for the treatment of non-neurological bleeding in animals has been published [13].

The major advantage of Phil is that complete embolization of target vessels can be achieved more quickly and safely and with less tissue toxicity compared to other embolic agents. Before the procedure, a careful evaluation of the indications and patient anatomy must be performed by the interventional radiologist, so that the common limits of the procedure, such as high flow conditions, can be overcome. Although three different viscosities of Phil (25%, 30% and 35%) are available, it can be safely injected in association with other materials such as coils or plugs to reduce the blood flow, thereby avoiding Phil misplacement and non-target embolization.

In our experience, pain occurred in patients where Phil was injected in the abdominal and peripheral areas and was related to the injection of DMSO. However, DMSO-related pain can be easily controlled with medical therapy. 

The iodine component covalently bonded to the co-polymer renders continuous mixing during injection unnecessary and provides Phil with a homogeneous radiopacity, maintaining the same visibility regardless of the length of the procedure. This feature allows a lower risk of microcatheter entrapment and a good visualization of the embolizing material in fluoroscopy, without causing artifacts on follow-up CT scans [15]. 

Entrapment of the microcatheter after embolization is a potential technical complication related to the use of the Phil; this difficulty was not found in our study, but it has been reported as a rare event by some authors in the neurological field, in which usually procedures last longer. A dedicated DMSO compatible detachable microcatheter with a tip relief region has been created to avoid microcatheter entrapment. With this microcatheter, by pulling away the microcatheter, with a moderate traction at the end of the procedure, the tip remains inside the injected Phil material in case of catheter entrapment. This possibility leads to three major advantages: the traction force exerted on the arteries is low, the risk of hemorrhage is minimized, and there is no risk of leaving the full catheter inside the body [11,22,23,24,25]. However, in our experience, no cases of microcatheter entrapment occurred; in order to avoid this complication, during or immediately after Phil injection, we pulled gently and very slowly backward the catheter’s tip. 

A frequent problem that occurred during our initial experience with Phil was represented by the partial or total occlusion of the catheter lumen, limiting the ability to perform subsequent angiography. This aspect is extremely important, particularly in cases were super-selective catheterization of vary small branches was achieved. To avoid this issue, after each Phil injection we inject again the DMSO to flush the dead-space of the microcatheter; this effectively allows to clean the catheter lumen for subsequent use in angiography or additional Phil injections, if needed. In our experience, the use of higher doses of DMSO did not cause complications, since no inflammatory reactions were observed, and pain was well controlled by medical therapy. Despite this, the operator should acknowledge the presence of residual embolic agent within the catheter death space prior to flushing with DMSO, in order to avoid non-target embolization. Alternatively, removal of the microcatheter can be safely performed under gentle aspiration and completion angiography performed through the angiographic catheter.

Other embolic agents with similar features to Phil are available on the market, particularly Onyx (Medtronic, Dublin, Ireland) and Squid (Emboflu, Gland, Switzerland) [26,27,28,29].

All these three embolizing agents are non-adhesive, in liquid form, in solution with DMSO, and therefore require DMSO-compatible catheters and microcatheters.

All these three embolic agents are available in different concentrations: In particular, in addition to the different Phil concentrations listed above, Onyx is available as Onyx18 (lower viscosity) or Onyx 34 (higher viscosity), while Squid is available as Squid12 (lower viscosity), Squid18 (standard viscosity), and Squid34 (higher viscosity); the different concentrations available for Phil have been described above [26,27,28,29].

Regarding the radiopacity and therefore the visibility of the embolic agent during the procedure, Phil is composed of a covalently bounded iodine component, while both Onyx and Squid are constituted of a mixture with tantalum powder that confers radiopacity to the embolic agent. This feature implies the need to shake the solution for about 20 min before use in order to suspend the tantalum powder, while Phil does not require this step and it is ready for use.

The presence of tantalum also gives greater radiopacity to the embolizing fluid, which ensures an excellent visualization during the procedure, but is often associated with beam-hardening artifacts in post-procedural CT scans. In order to reduce the beam hardening artifact, both Onyx and Squid are available with lower tantalum content than the current version, specifically named Onyx L and Squid LD [28,30].

Several articles showed that both Onyx and Squid are feasible alternatives for the treatment of abdominal and peripheral diseases [31,32], particularly a recent study demonstrated that Squid can be successfully used with a low complication rate in many abdominal diseases, showing a valid embolic action either combined with other embolic agents or alone. A recent systematic review showed that Onyx is a safe and effective embolic agent in treating acute hemorrhage outside of its customary neurovascular applications [18,19].

Limitations in the use of Phil^®^ for non-neurological indications include its relatively high cost, as compared with other embolic agents, and DMSO-related pain, especially when used in the abdomen.

Another aspect to consider is the need for dedicated catheters and DMSO-compatible microcatheters, which is necessary to avoid premature solidification inside the microcatheter. In addition, a cost-effectiveness evaluation study should be performed to completely validate its use in the peripheral area.

## 5. Conclusions

Our preliminary experience shows that Phil can be considered a safe and effective embolic agent for use, in the treatment of non-neurological acute and subacute bleeding including arterial and portal varices hemorrhages, achieving a high technical success in the management of emergency bleeding. In addition, Phil can safely be used in combination with other embolic agents such as coils, plugs, gelfoam, or other materials.

A cost-effectiveness evaluation study should be performed to completely validate its use in the peripheral area, and further studies will be needed to identify the most suitable indication for this device.

## Figures and Tables

**Figure 1 jcm-10-00701-f001:**
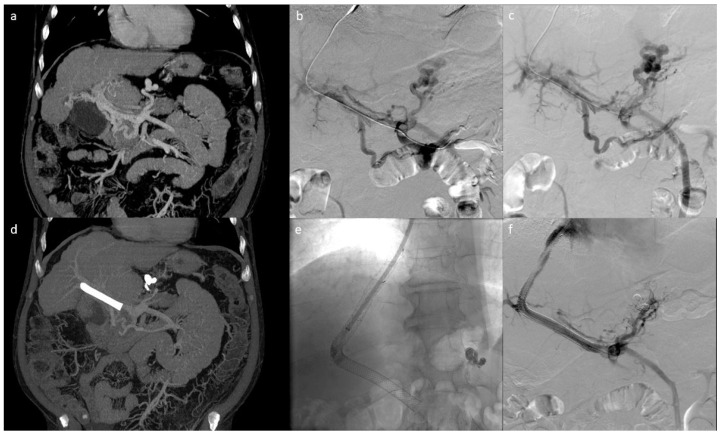
Patient with cavernomatosis and portal hypertension who had episodes of bleeding from gastric varices. (**a**) shows the pre-procedural CT scan with voluminous gastric varices. The patient underwent TIPS and embolization of gastric varices via jugular access. (**b**,**c**) show the pre-embolization portography; (**e**,**f**) show the post-embolization portography with TIPS placement; (**d**) shows the post-procedure follow-up CT demonstrating the correct embolization of the gastric varices in the absence of non-target embolizations).

**Figure 2 jcm-10-00701-f002:**
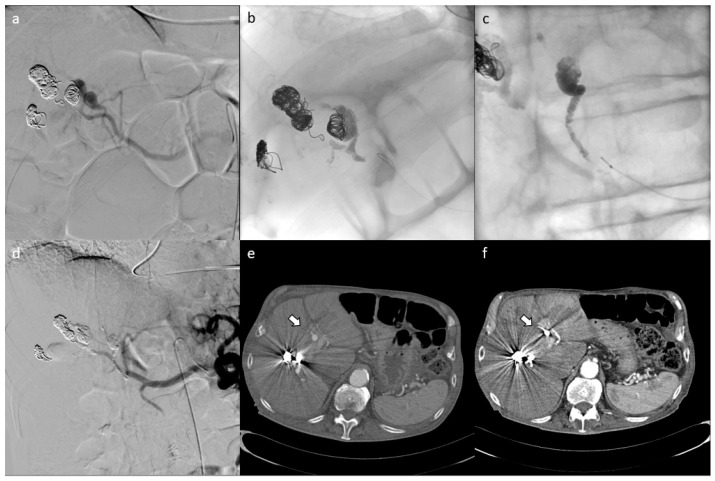
Patient with multiple post-pancreatitis hepatic pseudoaneurysms who had undergone previous embolization with metal coils and who presented with new hemoglobin drop underwent CT scan, demonstrating the presence of new pseudoaneurysms. (**e**) shows the pre-procedural CT scan with pseudoaneurysms and the results of the embolization with metal coils. The patient underwent embolization of the pseudoaneurysms ((**a**) shows pre-embolization angiography; (**b**–**d**) show outcomes of embolization and completion angiography). Post-procedural CT scan demonstrates the correct embolization of the pseudoaneurysms in the absence of non-target embolization (**f**).

## Data Availability

The data presented in this study are available on request from the corresponding author. The data are not publicly available due to privacy and ethical reasons.

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
