# Peer review of "Use of Phil Embolic Agent for Bleeding in Non-Neurological Interventions"

_jcm, 2021, doi:10.3390/jcm10040701_

Round 1

Reviewer 1 Report

Dear Editor, Dear Authors,

I was committed to review the paper "Use of Phill embolic agent for bleeding in non-neurological interventions", proposed for publication on Journal of Clinical Medicine.

The article deals with the clinical value (safety and efficacy) of a new embolic liquid agent in the hemorrhagic theatre. The use of liquid agents is gaining increasing attention in the embolization field and the matter is certainly of interest for endovascular specialists, mostly IR. Nevertheless, the work needs a wide major revision in terms of design study, scientific writing and eventually English. For instance the authors decide to address in Introduction the "abdominal bleeding", that is a wide clinical context definitely requiring, on my opinion, a more structured presentation. In M&M there is a lack of any definition for the study end-points (clinical success, angiographic success, etc.). Too many imprecisions:  "Phil system" is Phil embolic agent I guess; does it make sense to talk about injection rate of an embolic agent if, I guess, it was a manual injection? In scientific writing separate lines are not used. Conclusion is somehow hasty for a small clinical and such heterogenous series.

I would suggest that the authors focus on a careful revision of the literature before proceeding to rewrite the paper.

Author Response

Dear Reviewer,

thanks for the comments and observations, below you can find our modifications to the manuscript acording to your indications.   

  • "Nevertheless, the work needs a wide major revision in terms of design study, scientific writing and eventually English": thanks for the observetion, we enterely revise the english level of the draft that has undergone revision by an english mother tongue author.   
  • "For instance the authors decide to address in Introduction the "abdominal bleeding", that is a wide clinical context definitely requiring, on my opinion, a more structured presentation": thanks for the comment, unfortunately because the type of scenarios is very varied and for the brevity required by the introduction we decided not to explore this aspect in depth   
  • "In M&M there is a lack of any definition for the study end-points (clinical success, angiographic success, etc.)": thanks for the comment, we inserted this part in materials and methods (line 136-146) 
  • "Too many imprecisions:  "Phil system" is Phil embolic agent I guess; does it make sense to talk about injection rate of an embolic agent if, I guess, it was a manual injection?" thanks for the comment, the embolica agent was administered by a manual injection then we eliminated the injection rate (line 19 and line 95) 
  • "Conclusion is somehow hasty for a small clinical and such heterogenous series": thanks for the comment, we modified the conclusion section (line 295-297).  

Best regards,

L. Teodoli

Reviewer 2 Report

Dear Authors, this paper presents some interesting concerns, but there some criticisms I have raised.

The Author described a very interesting study on the use of Phill as embolic agent for bleeding in non-neurological interventions. It is first report on this field. The paper needs some corrections.

ABSTRACT

Line 24-25: Delete these considerations, you did not discuss in the abstract.

INTRODUCTION

Line 52-54: Revise this part.

Line 58-60: Move this part to Methods section.

Materials and Methods

First of all, it is fundamental to insert interval of time of this study.

Line 70: By..an interventional radiologist.

If are present, Please include exclusion criteria

Insert and clearly indicate key points and goal of this study.

RESULTS

PES: Specify first time you used this term

Last paragraph of the results presented considerations, insert it in discussion

DISCUSSION

When you compare Phil to others method it is preferable if you cite some reference and report percentage of success with others embolic agents.

Line 190: If you used others embolic agents in your patients after Phill, you should report it in Methods and discuss.

Line 200: Probably could be of interest insert mean time of procedures using Phil

Line 230: Probably could be of interest insert mean consume of Phill comparing with others agents for same procedures, in order to compare costs of procedures.

REFERENCES

Line 273: Verify this reference

Author Response

Dear Reviewer,

thanks for the comments and observations, below you can find our modifications to the manuscript according to your indications:

ABSTRACT "Line 24-25": thanks for the comment, we deleted these considerations.

INTRODUCTION

"Line 52-54": thanks for the observation, we changed this part.

"Line 58-60": thanks for the comment, we moved this part to materials and methods.

MATERIALS AND METHODS

"First of all, it is fundamental to insert interval of time of this study": thanks for the comment, we inserted the time of the study (line 70)

"Line 70: By..an interventional radiologist": thanks for the comment, we fixed it

"If are present, Please include exclusion criteria": We thanks the reviewer for this comment, the choice to utilize the Phill was left to the operator preference. Being this only a retrospective analysis is not possible to define an exclusion criteria. In order to give to the audience a wider point of view on the potential application of such a device in the emergency setting we introduced in the manuscript the numerosity of all bleeding patient referred to our IR unit in the same period (n=133 patients) comprehensive also of patients not treated with this device. This will give the reader a clearer idea of the role of this device. This part was addressed to line 70-71 in Materials and Methods.

"Insert and clearly indicate key points and goal of this study": thanks for the comment, we inserted the study end points section in materials and methods (line 136-146).

RESULTS

"PES: Specify first time you used this term": thanks for the comment, we added this part (line 140)

"Last paragraph of the results presented considerations, insert it in discussion": thanks for the comment we deleted this part from Results section and integrated it in Discussion section (line 279-281)  

DISCUSSION

"When you compare Phil to others method it is preferable if you cite some reference and report percentage of success with others embolic agents": thanks for the comment, we corrected this part

"Line 190: If you used others embolic agents in your patients after Phill, you should report it in Methods and discuss": thanks for the comment we inserted this informations in materials and methods (line 97-98) and in results (line 159-160).

"Line 200: Probably could be of interest insert mean time of procedures using Phil": thanks for the comment, we entered this information in materials and methods (line 99-100) and in results (line 164).

"Line 230: Probably could be of interest insert mean consume of Phill comparing with others agents for same procedures, in order to compare costs of procedures": thanks for the comment, we included the mean consume of Phil in results (line 156-158)

REFERENCES

"Line 273: Verify this reference": thank you for your comment, we have modified the reference and we have inserted the link to the official website of the product, inserted according to the guidelines of the newspaper; in that site is available the data sheet of the embolizing agent from which we have taken the information reported in the text of the article (line 333-334)

Best regards,

L. Teodoli

Round 2

Reviewer 1 Report

The manuscript has been certainly improved but still needs refinements.

« Neurological interventions » : I would use better « peripheral endovascular urgencies»

Same in introduction “outside the neurovascular area”, I would use “Peripheral applications”

I do not think that a separate paragraph “End points” is necessary, needs to integrate in the M&M

In Conclusions I would precise: Our preliminary experience

Institutional Review Board Statement: Usually retrospective observational studies do not need IRB. Authors were asked for this?

References: 15 references is not much even for an original article, authors need probably to extend the review of the literature to support the Introduction and their discussion.

Author Response

Dear Reviewer,

thanks for the comments and the contribution. I report below the modification of the manuscript according to your comments:

  • "« Neurological interventions »": Thanks for the comment; we think that underlying that our series was "non-neurological" instead of "peripheral endovascular" better delineate the pivotal strength of our study that firstly report Phil outside the neurological area.
  • “End points”: Thanks for the comment: we modified this paragraph and we inserted it in M&M section. 
  • "Our preliminary experience": Thanks for the comment, we modified this sentence (line 287).
  • "Institutional Review Board Statement" thanks for the comment, we have modified the sentence and left the statement about the Institutional Review Board we have obtained.
  • "References": thanks for the comment, we improved the references section

Best regards,

Dr. Leonardo Teodoli

Reviewer 2 Report

Dear Authors 

My compliments for improving this paper according to my concerns. 

Author Response

Dear Reviewer,

thanks for the contribution.

Best regards,

Dr. Leonardo Teodoli